# Effect of mupirocin for *Staphylococcus aureus* decolonization on the microbiome of the nose and throat in community and nursing home dwelling adults

Mary-Claire Roghmann[1,2]*, Alison D. Lydecker[2], Michelle Shardell[2,3], Robert T. DeBoy[2], J. Kristie Johnson[2,4], LiCheng Zhao[4], Lauren L. Hittle[3], Emmanuel F. Mongodin[3]

1 Geriatrics Research Education and Clinical Center, VA Maryland Health Care System, Baltimore, Maryland, United States of America, 2 Department of Epidemiology and Public Health, University of Maryland School of Medicine, Baltimore, Maryland, United States of America, 3 Department of Microbiology and Immunology and Institute for Genome Sciences, University of Maryland School of Medicine, Baltimore, Maryland, United States of America, 4 Department of Pathology, University of Maryland School of Medicine, Baltimore, Maryland, United States of America

* mroghmann@som.umaryland.edu

**Data Availability Statement:** Sequence data generated in this study have been deposited to Genbank and are linked to BioProject

## Abstract

### Objective

To characterize the microbial communities of the anterior nares (nose) and posterior pharynx (throat) of adults dwelling in the community and in nursing homes before and after treatment with intranasal mupirocin.

### Methods

*Staphylococcus aureus*-colonized adults were recruited from the community (n = 25) and from nursing homes (n = 7). *S. aureus* colonization was confirmed using cultures. Participants had specimens taken from nose and throat for *S. aureus* quantitation using quantitative PCR for the *nuc* gene and bacterial profiling using 16S rRNA gene sequencing over 12 weeks. After two baseline study visits 4 weeks apart, participants received intranasal mupirocin for 5 days with 3 further visits over a 8 week follow-up period.

### Results

We found a decrease in the absolute abundance of *S. aureus* in the nose for 8 weeks after mupirocin (1693 vs 141 fg/ul, p = 0.047). Mupirocin caused a statistically significant disruption in bacterial communities of the nose and throat after 1 week, which was no longer detected after 8 weeks. Bacterial community profiling demonstrated that there was a decrease in the relative abundance of *S. aureus* (8% vs 0.3%, p<0.01) 8 weeks after mupirocin and a transient decrease in the relative abundance of *Staphylococcus epidermidis* in the nose (21% vs 5%, p<0.01) 1 week after mupirocin.

PRJNA388722 in the NCBI BioProject database
(https://www.ncbi.nlm.nih.gov/bioproject/).

**Funding:** This work was supported by the United States (U.S.) Department of Veterans Affairs Clinical Sciences Research and Development Service (I01 CX000491-01A1) (MCR, JKJ, EFM); and the National Institute of Allergy and Infectious Diseases (1R01AI087865-01A1) (MCR, JKJ, EFM). The contents do not represent the views of the U.S. Department of Veterans Affairs or the United States Government. The content is solely the responsibility of the authors and does not necessarily represent the official views of the National Institute of Allergy and Infectious Diseases or the National Institutes of Health. The funders had no role in study design, data collection and interpretation, or the decision to submit the work for publication.

**Competing interests:** The authors have declared that no competing interests exist.

## Conclusions

Decolonization with mupirocin leads to a sustained effect on absolute and relative abundance of *S. aureus* but not for other bacteria in the nose. This demonstrates that a short course of mupirocin selectively decreases *S. aureus* in the nose for up to 8 weeks.

## Introduction

"Decolonization" is a rapidly growing strategy to prevent *Staphylococcus aureus* infections. Its use is fueled by healthcare policy initiatives, such as public reporting of healthcare associated infections which are often caused by *S. aureus* [1, 2]. Decolonization involves the application of antimicrobial agents such as mupirocin to the skin or mucosal surfaces. Mupirocin has high level of activity against staphylococci, streptococci and certain Gram-negative bacteria (*Haemophilus influenzae* and *Neisseria gonorrhoeae)* [3]. Staphylococci are abundant in the anterior nares [4, 5]. Streptococci are abundant in the throat [4].

Intranasal mupirocin, which is inadvertently swallowed, may alter the microbiota of the nose and throat. Given the role of the human microbiota as a barrier to infection, decolonization with mupirocin could have unintended negative consequences. For example, decolonization with intranasal mupirocin increases the risk of infections due to organisms other than *S. aureus*, including Gram-negative rods by 38% [6].

To assess the effect of mupirocin on the bacterial communities of the nose and throat, we conducted an interventional decolonization study. We have previously reported the results of our culture analysis specifically looking at S. aureus and pathogenic Gram-negative rods [7]. Here we present the results of our genomic analysis which allows us to look at bacterial communities as opposed to specific bacteria. To our knowledge, there are no other reports of the changes in bacterial communities of the nose and throat with intranasal mupirocin. Our objective was to compare the bacterial communities of the nose and throat in community and nursing home dwelling *S. aureus*-colonized adults before and after treatment with intranasal mupirocin. We hypothesized that treatment with mupirocin would decrease the abundance of staphylococci and streptococci, and conversely increase the abundance of *Enterobacteriaceae* within individuals.

## Material and methods

### Ethics

The study was conducted in accordance with the Declaration of Helsinki and national and international standards. The study was approved by the University of Maryland, Baltimore IRB and the VA Research and Development Committee at the VA Maryland Health Care System. The community dwelling portion of the study was approved on October 21, 2011 and the nursing home dwelling portion of the study was approved on April 28, 2014. Written informed consent was obtained from all study participants.

### Recruitment and study procedures

This was an interventional study in community and nursing home dwelling *S. aureus*-colonized adults. *S. aureus* colonization was confirmed with microbiological cultures. After two baseline study visits over 4 weeks, the nose and skin of participants were decolonized for 5 days with an 8 week follow-up period in which participants were seen at 1, 4 and 8 weeks after

mupirocin. The two study visits prior to decolonization allowed each participant to serve as their own control. In order to be included, participants needed to be *S. aureus* colonized on at least one of the following body sites during at least one of the baseline visits: anterior nares, throat, subclavian skin, femoral skin or perirectal skin. This convenience sample was recruited prospectively for this study from local VA primary care clinics and VA nursing homes and screened to document their eligibility and health status. Study visits took place in the General Clinical Research Center of the University of Maryland School of Medicine, and the nursing home units of Perry Point VA Medical Center and the Loch Raven VA Medical Center. Participant recruitment and follow-up took place between September 18, 2012 and September 22, 2015. Eligible participants were adults without: cancer treatment, HIV infection, immunosuppressive medications, nasal steroids, antibiotic (including chlorhexidine or mupirocin) use or recent hospitalization within 3 months. The nursing homes did not use mupirocin ointment as a part of infection control efforts. Participants who received antibiotics or were hospitalized during follow up were excluded. Non-invasive samples from the nose and throat were collected by research staff for genomic testing at each study visit [7]. After visit 2, participants received a 5-day course of nasal mupirocin ointment. Mupirocin 2% nasal ointment was applied by participants or nursing home staff twice a day. Neither the participants nor the nursing home staff were blinded to the mupirocin. Community dwelling participants filled out a subject diary. Nursing home dwelling participants had their regimen provided by and documented by nursing staff. Community dwelling participants received $50 per study visit. Nursing home dwelling participants received $5 per study visit. The final sample size was chosen based on feasibility and cost. There were no known protocol deviations during the study. The two studies underlying this manuscript are registered in ClinicalTrials.gov under study numbers NCT04218799 and NCT04222699. These studies were not registered in ClinicalTrials.gov prior to the start of participant enrollment as it was not required by the funders. The authors confirm that all ongoing and related trials for these drugs are registered.

## Microbiological methods

Enriched samples in tryptic soy broth with 6.5% NaCl and CHROMagar Staph aureus (Becton Dickenson; Sparks, MD) was used for the detection of *S. aureus* using standard microbiological procedures. Methicillin resistance was determined by oxacillin screen agar and antibiotic susceptibilities were performed following CLSI guidelines [8].

## Sample processing and DNA extraction

Total metagenomic DNA (mgDNA) was isolated as previously described [9, 10]. Negative extraction controls (PBS) were processed in parallel to each extraction to ensure no contaminating DNA was introduced during sample processing. All samples included in our analyses were negative for contaminating DNA.

## Quantitative PCR methods

Quantitative real-time PCR for total bacterial load quantitation was determined using 16S rRNA following methods by Walker et al [11] and *S. aureus* quantitation was determined following methods of Redel et al [12]. Briefly, total nucleic acid from nasal and throat samples were amplified and DNA concentration was determined using the BIO-RAD CFX96 Real-Time System. Standard curves were performed with each run consisted of seven 10-fold dilutions from 4 ng/ul to 4 fg/ul of *S. aureus* USA300.

## Microbiota profiling using 16S rRNA gene sequencing

Microbiota profiling was performed by sequencing, on Illumina HiSeq 2500 (Illumina, San Diego, CA) 2x300-bp PE, PCR amplicons of the V3-V4 hypervariable region of the 16S rRNA gene [13]. Previous studies have shown that this region is appropriate for species-level taxonomic assignments of the *Staphylococcus* genus [14–17]. Sample barcoding was performed using the dual-indexing strategy developed at the Institute for Genome Sciences [18], which allows sequencing of >1,500 samples in a single HiSeq 2500 run while providing high sequence coverage (~50,000 reads on average per sample) [13]. Briefly, PCR reactions were set-up using the 319F (ACTCCTACGGGAGGCAGCAG) and 806R (GGACTACHVGGGTWTCT AAT) 16S rRNA universal primers, each of which also included a linker sequence required for Illumina sequencing, and a 12-bp heterogeneity-spacer index sequence [18]. First step PCR amplifications were performed using Phusion High-Fidelity (Thermo Fisher, USA) and the following cycling parameters: 3 min at 95˚C, followed by 30 cycles of 30 sec at 95˚C, 30 sec at 58˚C, and 1 min at 72˚C, with a final step of 5 min at 72˚C [19]. Step-two PCR amplifications were set-up using custom barcode primers unique for each sample and 1 ul of diluted (1:20 dilution) first step products as template, under the following cycling parameters: 30 sec at 95˚C, followed by 10 cycles of 30 sec at 95˚C, 30 sec at 58˚C, 1 min at 72˚C, with a final step of 5 min at 72˚C. No-template negative controls were processed for each primer pair. The Sequal-Prep Normalization Plate kit (Life Technologies) was used for clean-up and normalization (25 ng of 16S PCR amplicon pooled for each sample) before sequencing.

## Data processing and statistical analyses of microbiome data

Following sequencing, 16S rRNA reads were initially screened for low-quality bases and short read lengths [18]. Paired-end read pairs were then assembled using PANDAseq [20] and the resulting consensus sequences were de-multiplexed (i.e. assigned to their original sample), trimmed of barcodes and primers, and assessed for chimeras using UCHIME [21] in *de novo* mode implemented in QIIME (v. 1.9.1) [22]. Quality-trimmed sequences were then clustered *de novo* into operational taxonomic units at 97% similarity cutoff using QIIME, and taxonomic assignments were performed using the RDP classifier implemented in QIIME and the Greengenes (v. 13.8) database as a reference.

The resulting taxonomic assignments were imported as a BIOM-formatted file into R (v. 3.3.2) using the RStudio (v. 1.0.44) integrated development environment (IDE), and processed/analyzed using the following R packages: Phyloseq (v. 1.19.1) [23], Vegan (v. 2.4–1) [24], and gpplot2 (v. 2.2.1) [25]. When appropriate, taxonomic assignments were normalized to account for uneven sampling depth with metagenomeSeq's cumulative sum scaling (CSS; implemented in R) [24, 26], a normalization method that has been shown to be less biased than the standard approach (total sum normalization). The Good's coverage index was calculated in R for each sample in order to ensure appropriate sequence coverage: samples with Good's coverage < 0.9 were discarded from the analyses. In addition, ultra-low abundant and likely-to-be spurious Operational Taxonomic Units (OTUs, <0.005% relative abundance and present in <10% of samples) were removed from the OTU table prior to the analyses described below.

Before normalization, within-sample comparisons using alpha-diversity measures were performed with the Observed estimator, as well as the Shannon diversity index, calculated using the Phyloseq R package. Because alpha-diversity metrics can be susceptible to uneven sampling depth between samples, alpha-diversity measures were compared after rarefaction to the minimum sampling depth of 2,000 sequences. The associations between participant dwelling and alpha diversity data were measured using the Wilcoxon rank sum test. Beta-diversity

(between-sample) comparisons were performed from CSS-normalized data through principle coordinates analysis (PCoA) plots of weighted UniFrac distance performed through the Phyloseq R package and tested for significance using the ANalysis Of SIMilarity (ANOSIM: 999 permutations) algorithm implemented in the Vegan package in R. Beta-diversity analyses did not use rarefied data.

Absolute abundance values of *S. aureus*, *S. epidermidis*, *Enterobacteriaceae* and *Streptococcaceae* were calculated by multiplying the normalized relative abundance values of each of these OTUs by the total bacterial counts assessed by 16SqPCR [5]. In this calculation, we did not take into consideration the number of 16S copies, as recent papers in the literature have suggested that 16S copies can only be accurately predicted for a limited fraction of known taxa, therefore potentially biasing calculations [27]. In addition and more importantly, variations in the number of 16S copies is generally small compared to variations in overall bacterial load [27]. Determination of statistically significant differences (adjusted p-value < 0.01) in OTU bacterial relative abundance between samples from different time points was performed using DESeq2 [28] implemented in R, which utilizes Benjamini-Hochberg multiple-inference correction. DESeq2 was used due to its high power in computing statistical significance of differentially-abundant features in high dimensional datasets derived from relatively small sample sizes. DESeq2 analyses did not use rarefied data.

The associations between participant dwelling and baseline characteristics were measured using t-tests for continuous variables and Fisher's exact tests for categorical variables. The associations between high and low S. aureus groups and abundance of *S. aureus*, *S. epidermidis*, *Enterobacteriaceae* in the nose over time were measured using the Wilcoxon rank sum test (Mann Whitney U test). The associations between abundance of *Streptococcaceae* in the throat over time within individuals were measured using the Wilcoxon signed ranks test. Unless otherwise specified, all statistical analyses were conducted with Stata 15 software (Stata Corporation, College Station, TX). As a sensitivity analysis, assessments over time were also analyzed using linear mixed models (LMM) with square-root transformations. For endpoints with excessive zeroes, zero-inflated gaussian linear mixed models were used. Analyses were performed with R statistical software version 3.6.3 using the nlme and NBZIMM packages.

## Accession number

Sequence data generated in this study have been deposited to Genbank and are linked to BioProject PRJNA388722 in the NCBI BioProject database (https://www.ncbi.nlm.nih.gov/bioproject/).

## Results

### Study population

Twenty participants were *S. aureus* colonized at both baseline study visits and 12 participants were colonized at one baseline visit. Baseline characteristics of our participants are shown in Table 1. There was no difference in *S. aureus* colonization at the anterior nares or throat by dwelling. Nine community dwelling participants and 3 nursing home dwelling participants (38% of all study participants) were colonized MRSA at baseline. All *S. aureus* isolates were susceptible to mupirocin. The nose sample for two participants at one visit could not be used. The nose and throat samples for three participants at one visit were missing. All 32 participants were adherent in taking mupirocin. There were no adverse events that were related to mupirocin or study procedures. Baseline characteristics did not differ between participants who were lost to follow-up and those retained. Participant flow is shown in Fig 1.

**Table 1. Baseline characteristics of participants with at least two study visits pre intervention and three study visits post intervention.**

| Characteristic | Community-based Participants (n = 25) | Nursing Home-based Participants (n = 7) | P value[*] |
|---|---|---|---|
| Age (years) (mean ± SD) | 47 ± 18 | 71 ± 14 | <0.01 |
| Male, n (%) | 15 (60) | 7 (100) | 0.07 |
| Race, n (%) | | | 0.04 |
| American Indian or Alaskan Native | 0 (0) | 1 (14) | |
| Asian | 1 (4) | 0 (0) | |
| African American | 15 (60) | 1 (14) | |
| White | 9 (36) | 5 (71) | |
| Body Mass Index (mean ± SD) | 26 ± 4 | 30 ± 4 | 0.05 |
| Diabetes, n (%) | 1 (4) | 1 (14) | 0.40 |
| Currently smoke, n (%) | 10 (40) | 0 (0) | 0.07 |
| *S. aureus* colonization[†] sites, n (%) | | | |
| Nose | 14 (56) | 5 (71) | 0.67 |
| Throat | 19 (76) | 4 (57) | 0.37 |
| Subclavian, femoral or perianal skin | 14 (56) | 2 (29) | 0.39 |

[*] p-values from t-tests for continuous variables and Fisher's exact tests for categorical variables

[†] As determined by culture

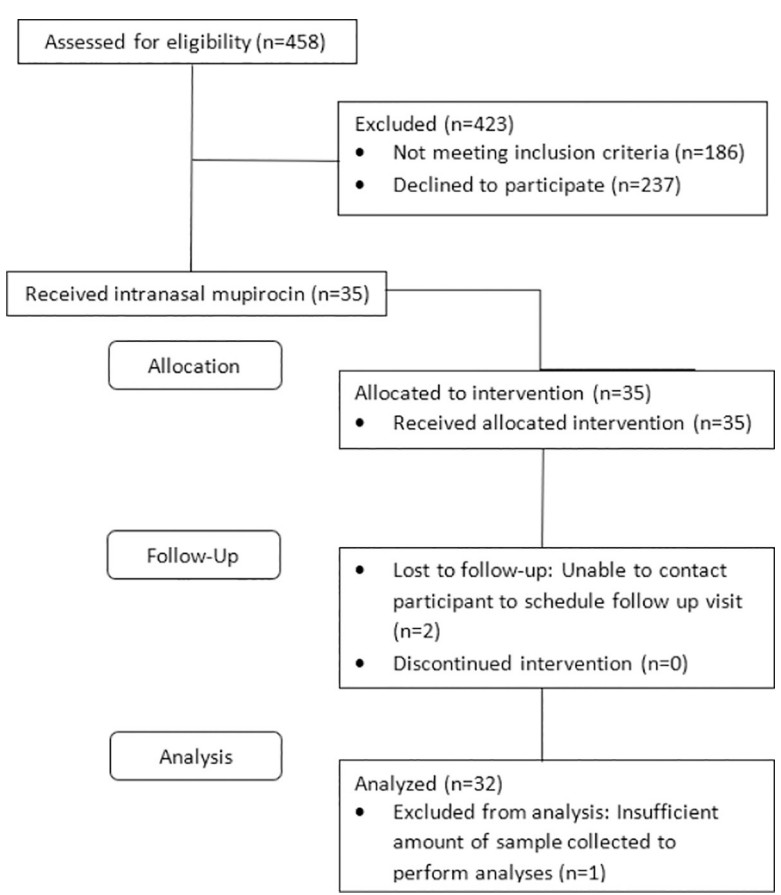

**Fig 1. CONSORT diagram depicting participant flow through the study.**

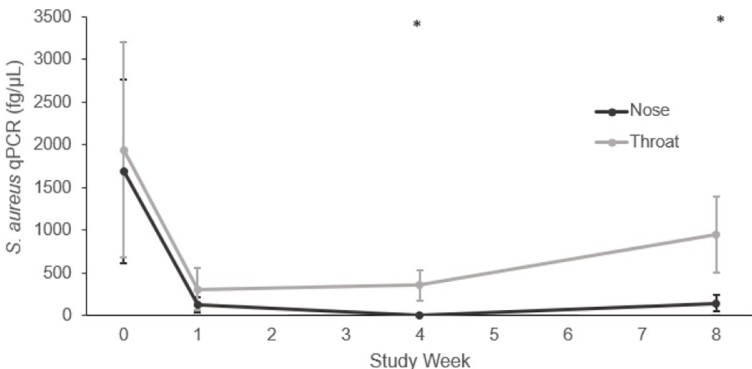

**Fig 2. Absolute abundance of *Staphylococcus aureus* as measured by qPCR over time by body site (fg/μL).** Data are presented as means ± SEMs. Statistical comparisons were made with the Week 0 time point within body site. Statistical significance was determined by zero-inflated Gaussian linear mixed models using a square root transformation. Asterisk, P<0.05.

## Absolute abundance of *S. aureus* and overall bacterial load assessed by qPCR

We found a statistically significant (*p* value = 0.011) sustained decrease in the absolute abundance of *S. aureus* in the nose for 8 weeks after mupirocin; the absolute abundance of *S. aureus* in the throat also decreased but not significantly (*p* value = 0.30) after mupirocin (mean *S. aureus* reduction: nose 1553 fg/μL, throat 993 fg/μL) (Fig 2). The overall decrease in the nose was driven by a subset of participants in the population with a high relative abundance of *S. aureus* in the nose defined by being in the upper quartile (S1 Fig). Because of this difference, we stratified by high *vs*. low relative abundance of *S. aureus* for nose samples in the remaining analyses. Eight participants had a high relative abundance of *S. aureus*, of which 3 (38%) were MRSA colonized on their nose swab at Week 0. Twenty-four participants had a low relative abundance of *S. aureus*, of which 5 (22%) were MRSA colonized on their nose swab at Week 0. There were no differences by dwelling (community *vs*. nursing home), sex (men *vs*. women) or methicillin resistance at study start (MRSA *vs*. MSSA).

The total bacterial load did not decrease in either the nose or throat with mupirocin; however, after 8 weeks, there was an increase in total bacterial load in both nose and throat which was statistically significant (*p* value <0.01) in the throat (Fig 3). There were no differences by dwelling, sex, or methicillin resistance at study start.

## 16S rRNA gene sequencing dataset

Bacterial community profiling using 16S rRNA gene sequencing and analysis was performed on a final (Good's coverage > 0.9) set of 312 samples. The final sequencing dataset contained a total of 20,427,872 16S rRNA gene sequences (65,473 sequences were obtained on average per sample, with a range of 2,015 to 407,309 sequences), representing 4,182 unique OTUs at a 97% similarity cut-off across all samples.

## Alpha diversity

Alpha diversity, a method to quantitate intra-sample diversity, was calculated using the Phyloseq package in R and reported using the Observed (total number of OTUs, a measure of community richness) and Shannon diversity indices. Because differences in sequence coverage—even close to sequence saturation—can have a significant impact on alpha diversity measures,

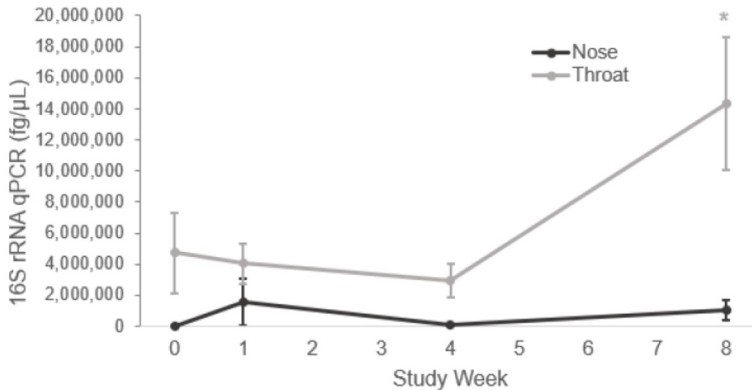

**Fig 3. Absolute abundance of overall bacterial load as measured by 16S rRNA qPCR over time by body site (fg/ μL).** Data are presented as means ± SEMs. Statistical comparisons were made with the Week 0 time point within body site. Statistical significance was determined by a Gaussian linear mixed model for the nose and a zero-inflated Gaussian linear mixed model for the throat. Both models used a square root transformation. Asterisk, P<0.05.

we calculated the Observed and Shannon indices on rarefied sequencing data (sampling depth: 2,000 sequences). There was an increase in observed OTU and Shannon diversity 1 week after mupirocin in those with a high relative abundance of *S. aureus* in their nose (observed OTU 78 vs 105, p = 0.03; Shannon 2.1 vs 2.6, p = 0.12) and no changes in the nose in those with a low relative abundance of *S. aureus* (observed OTU 93 vs 86, p = 0.26; Shannon 2.4 vs 2.2, p = 0.48) (Fig 4). There were no changes observed in the throat (S2 Fig).

## Beta diversity

Comparison of inter-sample diversity over time was characterized using beta-diversity analyses based on the weighted UniFrac distance. We did not detect any statistically significant sustained disruption in overall bacterial communities of the nose and throat as measured by change in weighted UniFrac distance over time. Fig 5 compares the weighted UniFrac distance between specimens from individual participants before mupirocin (Week -4 and Week 0) to the weighted UniFrac distance between specimens from participants before and after mupirocin (e.g. Week 1 and Week 0). In the nose, there was a sustained increase in weighted UniFrac distance in participants with a high relative abundance of *S. aureus* in the nose; however, this and the other increases between Week 1 and Week 0 were not statistically significant (Fig 5). There were similar patterns when stratified by dwelling and sex (S3 Fig).

A comparison of bacterial community structures between those with high *vs.* low relative abundance of *S. aureus* in the nose by body site showed that samples of participants with high relative abundance of *S. aureus* clustered with overlap to the participants with low relative abundance of *S. aureus* in the nose (Fig 6). Using the ANOSIM test of significance to compare all Week 0 and Week 1 clusters revealed significant differences in the nose (R = 0.1199 and p = 0.001) and throat (R = 0.07798 and p = 0.021). Significant differences were also observed for nose but not for throat when stratified by low relative abundance of *S. aureus*: nasal (R = 0.1218 and p = 0.004) and throat (R = 0.05613 and p = 0.051). No significant differences were observed when stratified by high relative abundance of *S. aureus*: nasal (R = 0.01897 and p = 0.348) and throat (R = 0.09375 and p = 0.133). However, reduced sample sizes caused by stratification might in part be responsible for these observations, masking the differences observed for all Week 0 and Week 1 samples. Comparison of all Week 0 and Week 8 clusters by site revealed no significant differences: nasal (R = 0.005484 and p = 0.337) and throat (R = -

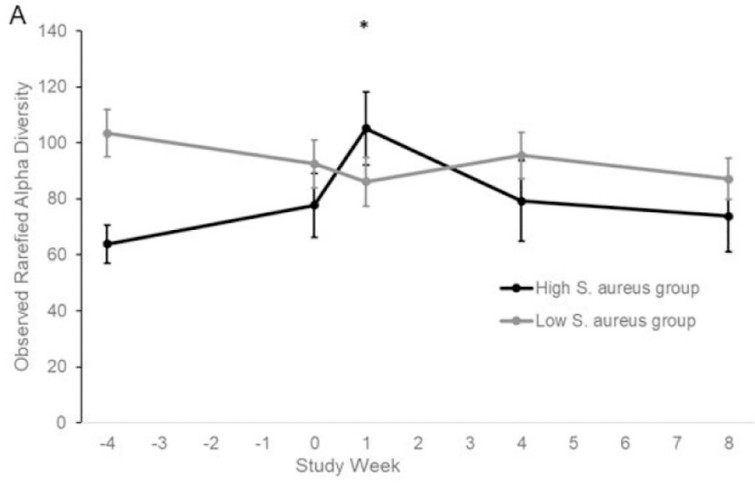

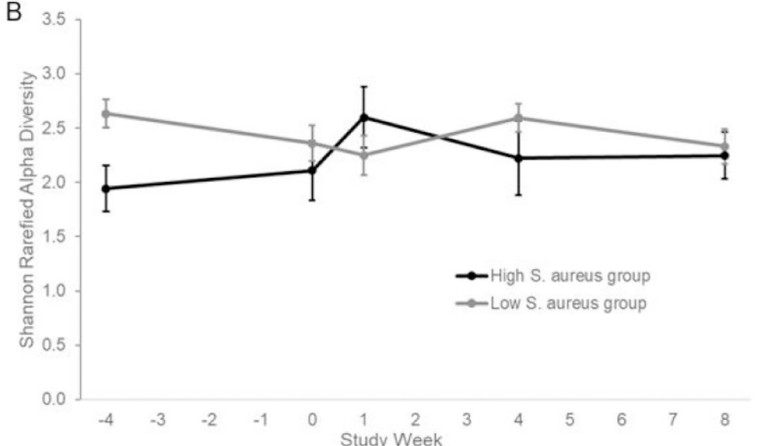

**Fig 4. Alpha diversity analyses of the nose samples over time at a sequencing depth of 2000 sequences per sample.** A. Observed Diversity Index, B. Shannon Diversity Index. Data are presented as means ± SEMs. Statistical comparisons were made between participants with high and low relative abundance of *Staphylococcus aureus* of the change from Week 0 to each time point. Statistical significance was determined by Gaussian linear mixed models. Asterisk, P<0.05.

0.01035 and p = 0.636). There were similar patterns when stratified by dwelling and sex (S4 Fig).

## Microbiome composition by body site and study population

Bacterial community profiling demonstrated that there was a sustained decrease in the relative and absolute abundances of *S. aureus* and a transient decrease in the relative and absolute abundances of *S. epidermidis* in the nose. There were no changes in the relative or absolute abundance of *Enterobacteriaceae*. There was a transient decrease in the relative and absolute abundance of *Streptococcaceae* in the throat. S5 Fig profiles the 15 most abundant taxa in nose and throat. Fig 7 shows the relative abundance over time of *S. aureus*, *S. epidermidis*, *Enterobacteriaceae* and *Streptococcaceae* before and after mupirocin. S6 Fig shows the absolute abundance as estimated by multiplying the relative abundance by the absolute burden of 16S rRNA over time of *S. aureus*, *S. epidermidis*, *Enterobacteriaceae* and *Streptococcaceae* before and after mupirocin. In addition to the top 15 taxa, DESeq2 was used to detect differential abundance of

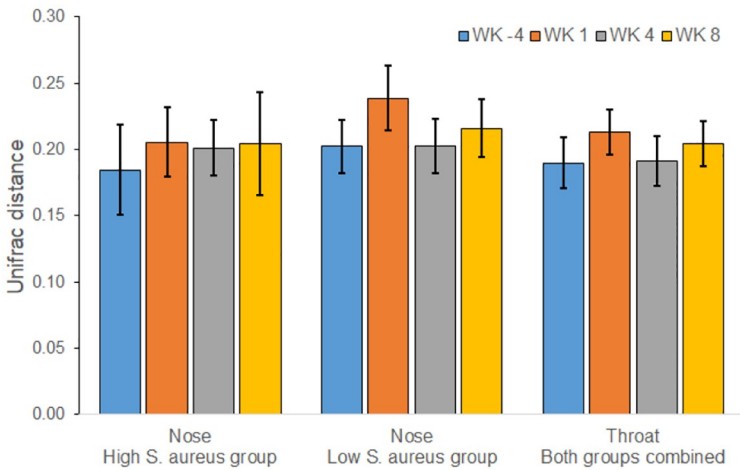

**Fig 5. Unifrac distances comparing distances to week 0 communities by body site.** Data are presented as means ± SEMs. Statistical comparisons were made between the distance of Week -4 to Week 0, and the distance of Week 0 to other time points within body site. Statistical significance was determined by Gaussian linear mixed models with a square root transformation. All tests were non-significant at the p<0.05 level.

less prevalent OTUs (S1 Table). Many of the same taxa decreased between Week 0 and 1, including OTUs of *S. aureus*, *S. epidermidis*, *S. haemolyticus*, unclassified *Corynebacterium* and *Propionibacterium* in the nose. At week 8, only OTUs of *S. aureus* and *Haemophilus* were differentially abundant in the nose. In the throat, comparing Week 0 and 1, DESeq2 identified decreases of *S. epidermidis*, unclassified *Streptococci* and *Gemellaceae* (formerly a *Neisseria*). At week 8, only an OTU of *S. aureus* was differentially abundant in the throat.

## Discussion

After mupirocin, we found a statistically significant decrease in the absolute and relative abundance of the *S. aureus* in the nose for 8 weeks. There was an increase in absolute abundance of overall bacterial load in both nose and throat at 8 weeks. Mupirocin caused a statistically significant transient disruption in bacterial communities of the nose and throat as measured by weighted Unifrac distances, which was no longer detected after 8 weeks. Bacterial community profiling demonstrated that there was a sustained decrease in the relative abundance of *S. aureus* and a transient decrease in the relative abundance of *S. epidermidis* in the nose. There was a transient decrease in the relative abundance of unclassified *Streptococcus* in the throat. In addition, there were transient decreases in OTU at a lower abundance: other coagulase positive staphylococci, *Propionibacterium*, *Corynebacterium*, and *Gemellaceae* and sustained decrease in *Haemophilus*.

The sustained decrease we observed of *S. aureus* in the nose following decolonization is consistent with culture based studies of mupirocin in community dwelling adults [29–31] and nursing home residents [32]. Of note, intranasal mupirocin did not increase colonization with Gram-negative pathogens in the nose or throat suggesting that any shift in infections from Gram-positive to Gram negative pathogen did not occur due to colonization shifts.

After mupirocin, there was an increase in absolute abundance of total bacteria load as measured by 16S rRNA qPCR in both nose and throat at 8 weeks. This was an unexpected result; we repeated a sample of the specimens and replicated our results. Typically, antibiotics, which are often broad spectrum, will decrease the overall burden of bacterial DNA [33, 34]. For example, sinus aspirates from adults with acute exacerbations of chronic rhinosinusitis

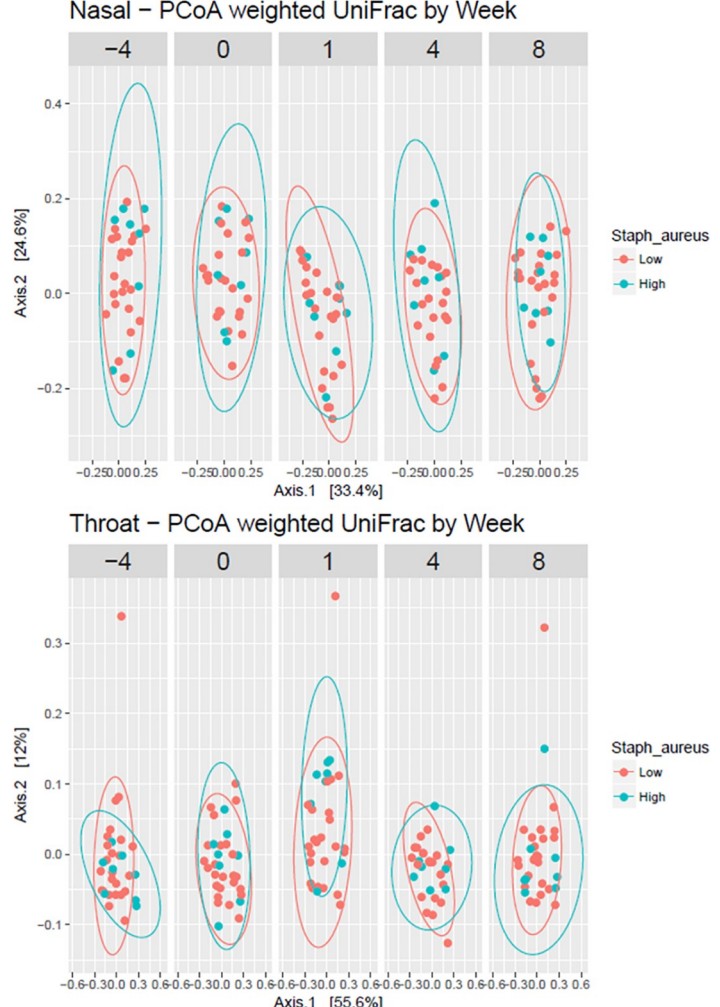

**Fig 6. Principal coordinates (PCs) analysis of beta diversity metrics by body site, showing distances from unweighted Unifrac over time.** The ellipses represent 95% confidence intervals for clustered specimens from participants with high relative abundance of *S. aureus* in the nose vs patients with low relative abundance of *S. aureus* in the nose. ANOSIM Test of Significance comparing week 1 to week 0: Nose R = 0.1199 and p = 0.001; Throat R = 0.07798 and p = 0.021.

decreased by several logs after levofloxacin [34]. Hauser and colleagues report an increase in bacterial burden in the sinuses two weeks after endoscopic sinus surgery and amoxicillin-cla-vulanate which then decreased to a level similar to the time of surgery [35]. Mupirocin is a narrow spectrum antibiotic compared with most antibiotics. It is possible that a transient decrease in staphylococci or streptococci leads to changes in the bacterial community which promote growth.

The changes seen with bacterial community profiling after mupirocin are consistent with what is known about the bacterial susceptibility to mupirocin. Mupirocin shows a high level of activity against staphylococci and streptococci and against certain Gram-negative bacteria; it is much less active against most Gram-negative bacilli and anaerobes [3]. We saw transient decreases in coagulase-negative staphylococci in the nose and streptococci in the throat. These relatively muted changes in response to mupirocin were also observed by Grice and colleagues in the skin microbiota of a mouse model [36]. In contrast to the transient changes in

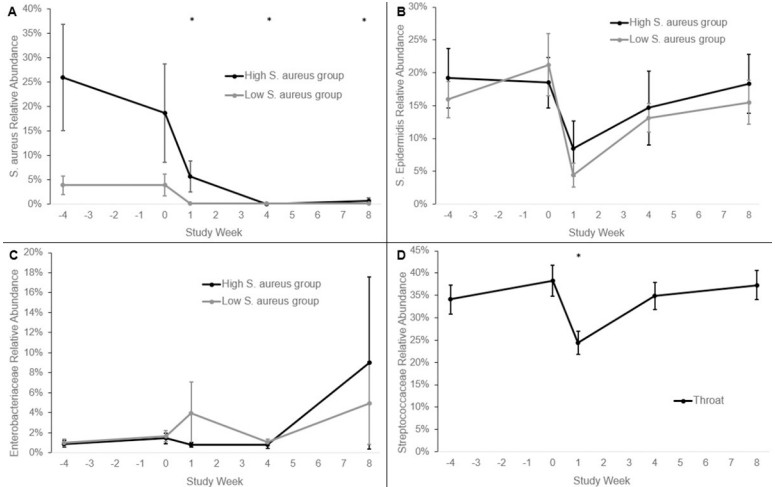

**Fig 7. Relative abundance of staphylococci, streptococci, and Enterobacteriaceae over time.** Data are presented as means ± SEMs. Statistical comparisons were made between participants with high and low relative abundance of *Staphylococcus aureus* of the change from Week 0 to each time point within the nose (A, B, C) or of the change from Week 0 to each time point within the throat (D). Statistical significance was determined by Gaussian linear mixed models or zero-inflated Gaussian linear mixed models as appropriate. Square root transformations were performed as necessary. Asterisk, P<0.05.

coagulase-negative staphylococci, we saw a sustained decrease in *S. aureus* in the nose. It is possible that the regrowth of coagulase-negative staphylococci in the nose lead to long term suppression of *S. aureus* as coagulase-negative staphylococci species have been identified as natural competitors of *S. aureus* [37–39].

Mupirocin caused a statistically significant initial disruption in bacterial communities of the nose and throat as measured by change in weighted Unifrac distances which was not sustained after 8 weeks. The disruption in the nose of those participants with a high relative abundance of S. aureus, although not statistically significant, was maintained post-mupirocin treatment, in contrast to participants in the low S. aureus group where oscillations in bacterial community structure could be observed. This is expected since there was a sustained decrease in *S. aureus* relative abundance after mupirocin. There was also a transient increase in alpha diversity indices in this subset 1 week after mupirocin. The sub-population with a high relative abundance of *S. aureus* in the noses appears most affected by the use of mupirocin.

Our study is limited by a small sample size particularly of nursing home residents. It is possible that studying a larger or more specific population, for example, adults with *S. aureus* as the dominant taxa in the bacterial community of the nose, might demonstrate sustained disruption of the microbiota after intranasal mupirocin. We are also limited by our follow-up of 8 weeks. We cannot comment on whether *S. aureus* re-enters the nasal microbiome after a longer time period. To our knowledge, this is the first report of the ecologic effect of a decolonization regimen in patient populations.

Decolonization with mupirocin leads to a sustained effect on *S. aureus* absolute and relative abundance in the nose consistent with past clinical trials of mupirocin. It is intriguing that mupirocin's long term effect is predominantly on *S. aureus* despite high susceptibility of other staphylococci and streptococci. Our study demonstrates that a short course of mupirocin selectively decreases *S. aureus* in the nose for up to 8 weeks sparing other bacteria. This selective antimicrobial action makes mupirocin an excellent option for short-term decolonization of *S. aureus*. In an era of increasing emphasis on the importance of choosing antibiotics with a

narrow spectrum, our report serves as an example of the type of approach which needs to be examined for other antibiotics.

## Supporting information

**S1 Checklist.**
(DOCX)

**S1 Fig. Absolute abundance of *Staphylococcus aureus* as measured by qPCR over time in the nose of participants with a high vs. low relative abundance of *S. aureus.*** Data are presented as means ± SEMs. Statistical comparisons were made with the Week 0 time point within SA group. Statistical significance was determined by Wilcoxon signed ranks test. Asterisk, $P < 0.05$.
(TIF)

**S2 Fig. Alpha diversity analyses of the throat samples over time at a sequencing depth of 2000 sequences per sample.** A. Observed Diversity Index, B. Shannon Diversity Index. Data are presented as means ± SEMs. Statistical comparisons were made with the Week 0 time point. Statistical significance was determined by Wilcoxon rank sum test (Mann Whitney U test). Asterisk, $P < 0.05$.
(TIF)

**S3 Fig.** Unifrac distances comparing distances to week 0 communities by body site stratified by dwelling (A: nose, B: throat) and gender (C: nose, D: throat). Data are presented as means ± SEMs. Statistical comparisons were made between the distance of Week -4 to Week 0, and the distance of Week 0 to other time points within body site. Statistical significance was determined by Wilcoxon signed ranks test. Asterisk, $P < 0.05$.
(TIF)

**S4 Fig.** Principal coordinates (PCs) analysis of beta diversity metrics by body site stratified by dwelling (A) and gender (B), showing distances from unweighted Unifrac over time. The ellipses represent 95% confidence intervals for clustered specimens from participants by gender and dwelling.
(TIF)

**S5 Fig. Distribution of the top 15 bacterial taxa at the lowest taxonomic classification for each body site over time stratified by the relative abundance of *S. aureus* in the nose.** Error bars show standard errors. A. Nose, B. Throat.
(TIF)

**S6 Fig. Absolute abundance of staphylococci, streptococci, and Enterobacteriaceae over time.** Data are presented as means ± SEMs. Statistical comparisons were made between participants with high and low relative abundance of *Staphylococcus aureus* of the change from Week 0 to each time point within the nose (A, B, C) or of the change from Week 0 to each time point within the throat (D). Statistical significance was determined by Wilcoxon rank sum test (Mann Whitney U test) (A, B, C) or the Wilcoxon signed ranks test (D). Asterisk, $P < 0.05$.
(TIF)

**S1 Table. Differentially abundant bacteria in the nose and throat over time using generalized linear models of abundance based on the negative binomial distribution [1].**
(DOCX)

**S1 File.**
(PDF)

**S2 File.**
(PDF)

## Acknowledgments

This work supported by the University of Maryland Institute for Clinical Translational Science and the University of Maryland General Clinical Research Center.

## Author Contributions

**Conceptualization:** Mary-Claire Roghmann, Emmanuel F. Mongodin.

**Data curation:** Alison D. Lydecker, Emmanuel F. Mongodin.

**Formal analysis:** Alison D. Lydecker, Michelle Shardell, Robert T. DeBoy, Emmanuel F. Mongodin.

**Funding acquisition:** Mary-Claire Roghmann.

**Investigation:** J. Kristie Johnson, LiCheng Zhao, Lauren L. Hittle.

**Methodology:** Mary-Claire Roghmann, J. Kristie Johnson, Emmanuel F. Mongodin.

**Project administration:** Mary-Claire Roghmann.

**Supervision:** Mary-Claire Roghmann.

**Writing – original draft:** Mary-Claire Roghmann, Emmanuel F. Mongodin.

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
