## [Decision Letter · Decision Letter 0]

15 Apr 2020

PONE-D-19-32171

Effect of mupirocin for Staphylococcus aureus decolonization on the microbiome of the nose and throat in community and nursing home dwelling adults

PLOS ONE

Dear Dr. Roghmann,

Thank you for submitting your manuscript to PLOS ONE. After careful consideration, we feel that it has merit but does not fully meet PLOS ONE’s publication criteria as it currently stands. Therefore, we invite you to submit a revised version of the manuscript that addresses the points raised during the review process.

The manuscript has been assessed by two reviewers; their comments are available below.

The reviewers have raised some concerns about the study which need attention in a revision. The reviewers request clarification on the definition of carriage employed and note that information should be provided on how DNA detection results correspond to culture results. The reviewers also raise concerns about the approach to the statistical analyses and recommend a re-analysis using different statistical methodology.

Could you please revise the manuscript to carefully address the concerns raised by the reviewers?

We would appreciate receiving your revised manuscript by May 29 2020 11:59PM. To enhance the reproducibility of your results, we recommend that if applicable you deposit your laboratory protocols in protocols.io, where a protocol can be assigned its own identifier (DOI) such that it can be cited independently in the future. For instructions see: http://journals.plos.org/plosone/s/submission-guidelines#loc-laboratory-protocols

We look forward to receiving your revised manuscript.

Kind regards,

Iratxe Puebla

Deputy Editor-in-Chief, PLOS ONE

Journal Requirements:

2. Thank you for submitting your clinical trial to PLOS ONE and for providing the name of the registry and the registration number. The information in the registry entry suggests that your trial was registered after patient recruitment began. PLOS ONE strongly encourages authors to register all trials before recruiting the first participant in a study.

a) your reasons for your delay in registering this study (after enrolment of participants started);

b) confirmation that all related trials are registered by stating: “The authors confirm that all ongoing and related trials for this drug/intervention are registered”.

Please also ensure you report the date at which the ethics committee approved the study as well as the complete date range for patient recruitment and follow-up in the Methods section of your manuscript.

Reviewers' comments:

Reviewer's Responses to Questions

**Comments to the Author**

1. Is the manuscript technically sound, and do the data support the conclusions?

Reviewer #1: Yes

Reviewer #2: No

2. Has the statistical analysis been performed appropriately and rigorously? 

Reviewer #1: I Don't Know

Reviewer #2: No

3. Have the authors made all data underlying the findings in their manuscript fully available?

Reviewer #1: Yes

Reviewer #2: Yes

4. Is the manuscript presented in an intelligible fashion and written in standard English?

Reviewer #1: Yes

Reviewer #2: Yes

5. Review Comments to the Author

Reviewer #1: This paper presents the genomic analysis of an interventional before and after trial of decolonisation with mupirocin in adults from the community and nursing homes. This is a follow up report and the study was earlier reported with culture results, showing sustained decrease in S. aureus by culture, but no increase in GNR colonisation. Using quantification of total and S. aureus DNA, as well as microbiome analysis by 16s rDNA sequencing, the authors establish that a sustained effect on S. aureus abundance is specifically shown in a group of 33 adults for up to 8 weeks after topical mupirocin.

The topic is an important one; unintended consequences of decolonisation must be weighed against the possible protective effects. Introduction provides a clear rationale and context for the study, and includes a tested hypothesis (that decrease in staphylococci and streptococci, with a reciprocal increase in the abundance of Enterobacteriaceae would be seen following decolonisation). There are some aspects of the methods I would like to see more clearly explained, but overall the data presented support their conclusions. I would suggest minor revisions to clarify these.

Major comments

- Colonisation was determined by two visits over 4 weeks. As there is evidence of frequent loss and gain events in S. aureus carriage (especially in younger people), I think some more detail on what definition of carriage was applied is needed. In particular, what number of sites over how many visits had to yield S. aureus to qualify as a carrier. The risk with using a single positive swab as evidence of carriage is that transient carriage is frequently observed, and the subsequent decrease in S. aureus carriage may reflect natural fluctuations, rather than the effect of topical antibiotics. This would explain the lack of sustained effect on other Gram positive organisms seen with mupirocin.

- The remaineder of inclusion and exclusion criteria are well presented. The exclusion criteria are sensible and not overly restrictive. Intervention was 5 days of bd nasal mupirocin, administered by subjects or the staff for NH patients, which appears reliable even if not directly observed by study staff. A convenience sampling strategy and sample size were employed.

- The Microbiological methods are clearly stated

- Genomics included appropriate negative controls in each extraction

- The metagenomics methods were reasonably clear to me (though I am not a metagenomics expert), and consideration has been given to the issues of bias introduction by PCR amplification, and includes consideration of a pre-specified, biologically based hypothesis.

- It is not clear to me why while S. aureus, S. epidermidis and Enterobacteriacae in nose and throat were examined, Streptococcaeceae were not examined in the nose. While naturally more abundant in the throat, these can certainly from part of the nasal microbiome (https://openres.ersjournals.com/content/4/4/00066-2018) . The exclusion should be explained.

Results

- Baseline characteristics are clearly presented, a flow diagram of recruitment and analysis is presented. It is notable that nearly half the community participants did not have S. aureus cultured from nose prior to treatment. I would like to see details of how S. aureus DNA detection correspond with culture results. The patients have been stratified by relative abundance of DNA (relative to this small cohort), but if culture detection is a better explanation than this parameter, this should be presented.

- Relative abundance of S. aureus DNA was decreased following mupirocin in nose but not throat; note that the nasal abundance is only just statistically significant by a nominal alpha (eg 0.05); I would suggest some quantitative presentation of the results to allow reader to determine if they think this decrease is quantitatively significant.

- Other interesting results include:

An increase in total increase in bacterial load in the throat at 8 weeks (more than 3-fold, and statistically significant) after mupirocin;

An increase in diversity indices at 1 week in the nose, only, and only for high SA abundance

No significant sustained disruption of the communities as measured by uniFrac distance

- Differences in bacterial community structures at week 0 and 1 that were not sustained over time. It is notable these differences were significant in the group as a whole, and in the low abundance stratum, but not the high abundance stratum. This is rather the opposite of what I might expect given that indices of diversity within samples were only significant in the high S. aureus group. While the authors comment this may be due to smaller group sizes, they do not explore other interpretations, and I would like to see comment on how these findings can be integrated, or whether one or both findings may be chance observations.

Discussion

- Overall, the discussion presents a balanced synthesis of the results and is clearly written.

- Absolute and relative abundance of S. aureus declines in the nose after mupirocin

- The unexpected finding of increased bacterial abundance after mupirocin is explored and possible explanations proposed; appropriately the authors retain some uncertainty about the weight of this finding.

- In Line 337-338 the authors state “There was a sustained, although not statistically significant, disruption in the nose of those participants with a high relative abundance of S. aureus”. Looking at 5A, and the overlapping confidence intervals of the UniFrac distance, and the larger (but still not significant) disruption seen in the low abundance group, this data does not strongly support this statement, and I suggest some moderation of the findings here.

Minor comments

- The abbreviation OTU (first used line 148) is not defined in the manuscript

Reviewer #2: Thirty-two adults with staphylococcus aureas colonization were included in the study. Nose and throat specimens were collected for quantitative PCR and bacterial profiling over a 12-week period. All participants received intranasal mupirocin treatment. The conclusions are unclear.

Major revision:

Statistical methods: A mixed linear model testing for changes over time would be superior to testing for differences by repeatedly applying Wilcoxon rank sum tests. The model offers flexibility for testing several factors simultaneously. Additionally, the model can adjust for variables that were significantly different between nursing home patients and community patients as indicated in Table 1.

Minor revision:

Table 1: Indicate the statistical testing methods used to calculate the p-values.

6. PLOS authors have the option to publish the peer review history of their article (what does this mean?). If published, this will include your full peer review and any attached files.

Reviewer #1: No

Reviewer #2: No

---

## [Author Response · Author response to Decision Letter 0]

12 Oct 2020

Thank you for the reminder about PLOS ONE’s style requirements. These formatting changes have been made.

2. Thank you for submitting your clinical trial to PLOS ONE and for providing the name of the registry and the registration number. The information in the registry entry suggests that your trial was registered after patient recruitment began. PLOS ONE strongly encourages authors to register all trials before recruiting the first participant in a study.

a) your reasons for your delay in registering this study (after enrolment of participants started);

b) confirmation that all related trials are registered by stating: “The authors confirm that all ongoing and related trials for this drug/intervention are registered”.

Please also ensure you report the date at which the ethics committee approved the study as well as the complete date range for patient recruitment and follow-up in the Methods section of your manuscript.

Please see lines 100-102 for our reason for delay in registering the studies and confirmation that all related trials are registered. The dates of ethics committee approval have been added in lines 71-72. The complete date range for patient recruitment and follow-up is found on lines 85-86.

Thank you for bringing this to our attention. We have removed the phrase as all the data is publicly available in Genbank.

5. Review Comments to the Author

Reviewer #1: This paper presents the genomic analysis of an interventional before and after trial of decolonisation with mupirocin in adults from the community and nursing homes. This is a follow up report and the study was earlier reported with culture results, showing sustained decrease in S. aureus by culture, but no increase in GNR colonisation. Using quantification of total and S. aureus DNA, as well as microbiome analysis by 16s rDNA sequencing, the authors establish that a sustained effect on S. aureus abundance is specifically shown in a group of 33 adults for up to 8 weeks after topical mupirocin.

The topic is an important one; unintended consequences of decolonisation must be weighed against the possible protective effects. Introduction provides a clear rationale and context for the study and includes a tested hypothesis (that decrease in staphylococci and streptococci, with a reciprocal increase in the abundance of Enterobacteriaceae would be seen following decolonisation). There are some aspects of the methods I would like to see more clearly explained, but overall the data presented support their conclusions. I would suggest minor revisions to clarify these.

Major comments

- Colonisation was determined by two visits over 4 weeks. As there is evidence of frequent loss and gain events in S. aureus carriage (especially in younger people), I think some more detail on what definition of carriage was applied is needed. In particular, what number of sites over how many visits had to yield S. aureus to qualify as a carrier. The risk with using a single positive swab as evidence of carriage is that transient carriage is frequently observed, and the subsequent decrease in S. aureus carriage may reflect natural fluctuations, rather than the effect of topical antibiotics. This would explain the lack of sustained effect on other Gram positive organisms seen with mupirocin.

In order to be included, participants needed to be S. aureus colonized on at least one of the following body sites during at least one of the baseline visits: anterior nares, throat, subclavian skin, femoral skin or perirectal skin. Twenty participants were S. aureus colonized at both baseline study visits and 12 participants were colonized at one baseline visit. This has been added to the manuscript in lines 79-81 and 191-192. Our stratification by relative abundance of S. aureus is done to distinguish between transient and persistent carriage, since those with persistent carriage have a greater abundance of S. aureus (1). Since the distinction between transient and persistent carriers is not made clinically and people with either are decolonized with mupirocin, our eligibility criteria are clinically relevant.

- The remainder of inclusion and exclusion criteria are well presented. The exclusion criteria are sensible and not overly restrictive. Intervention was 5 days of bd nasal mupirocin, administered by subjects or the staff for NH patients, which appears reliable even if not directly observed by study staff. A convenience sampling strategy and sample size were employed.

- The Microbiological methods are clearly stated

- Genomics included appropriate negative controls in each extraction

- The metagenomics methods were reasonably clear to me (though I am not a metagenomics expert), and consideration has been given to the issues of bias introduction by PCR amplification, and includes consideration of a pre-specified, biologically based hypothesis.

- It is not clear to me why while S. aureus, S. epidermidis and Enterobacteriacae in nose and throat were examined, Streptococcaeceae were not examined in the nose. While naturally more abundant in the throat, these can certainly from part of the nasal microbiome (https://openres.ersjournals.com/content/4/4/00066-2018) . The exclusion should be explained.

This was a specimen from the anterior nares which is more like the skin than the nasopharynx and Streptococcaeceae are not abundant in the anterior nares. This dataset included the top 15 bacteria that were found in the nose. The median relative abundance in the nose for Streptococcaeceae is 1.0% across all visits (mean of 3.2%). This can also be seen on panel A of Figure S5. There was not sufficient abundance in the nose for us to include it in the analysis.

Results

- Baseline characteristics are clearly presented, a flow diagram of recruitment and analysis is presented. It is notable that nearly half the community participants did not have S. aureus cultured from nose prior to treatment. I would like to see details of how S. aureus DNA detection correspond with culture results. The patients have been stratified by relative abundance of DNA (relative to this small cohort), but if culture detection is a better explanation than this parameter, this should be presented.

The objective of this study was not to compare DNA sequencing-based vs culture-based detection of S. aureus. These two approaches are fundamentally different and not easily comparable (cultivation involved broth enrichment whereas sequencing did not; cultivation detects metabolically-active bacteria that can grow in culture medium, whereas DNA-based sequencing can detect live, dormant or even dead bacterial cells, etc). The use of relative abundance was intended to distinguish between persistent vs intermittent or transient colonization. Thus we do not present this comparison.

- Relative abundance of S. aureus DNA was decreased following mupirocin in nose but not throat; note that the nasal abundance is only just statistically significant by a nominal alpha (eg 0.05); I would suggest some quantitative presentation of the results to allow reader to determine if they think this decrease is quantitatively significant.

Thank you for mentioning this. The mean reduction in absolute abundance of S. aureus from Week 0 to Week 8 was 1553 fg/µL in the nose and 993 fg/µL in the throat. This has been added to the manuscript in line 233.

- Other interesting results include:

An increase in total increase in bacterial load in the throat at 8 weeks (more than 3-fold, and statistically significant) after mupirocin;

An increase in diversity indices at 1 week in the nose, only, and only for high SA abundance

No significant sustained disruption of the communities as measured by uniFrac distance

- Differences in bacterial community structures at week 0 and 1 that were not sustained over time. It is notable these differences were significant in the group as a whole, and in the low abundance stratum, but not the high abundance stratum. This is rather the opposite of what I might expect given that indices of diversity within samples were only significant in the high S. aureus group. While the authors comment this may be due to smaller group sizes, they do not explore other interpretations, and I would like to see comment on how these findings can be integrated, or whether one or both findings may be chance observations.

We believe the reviewer refers to the Weighted UniFrac distances (“bacterial community structures”) from the beta-diversity analyses in Figs 5A/B and S4, and Observed and Shannon alpha-diversity indices (“indices of diversity within samples”) in Fig. 4A. These are 2 different and independent approaches aimed to characterize different features of complex bacterial communities. Our alpha-diversity analysis showed an increase in Observed species (measure of community richness) and Shannon index (measure of community diversity) in the high S. aureus group (but not in the low S. aureus group) between week 0 and 1. This is expected, given that when mupirocin is applied and clears S. aureus (and generally other Gram-positive bacteria), this provides an opportunity for other bacteria not affected by mupirocin to take over the environmental niche previously occupied by S. aureus. As S. aureus colonizes the anterior nares again after the mupirocin treatment is over, there is an overall decrease of bacterial diversity as S. aureus outcompetes other bacteria to become dominant again (Fig. 3).

Beta diversity compares bacterial community structure between samples, which is not directly related to alpha-diversity, since it measures similarity (or dissimilarity) between pairs of samples. The metric we used in our analyses (weighted UniFrac) compares similarity between bacterial communities, while also taking into consideration the phylogenetic relatedness of bacterial community members in the pair of samples compared, as well as their relative abundance. Therefore, it is possible that the increase in the Observed species richness (alpha-diversity) between week 0 and 1 seen for the high S. aureus group might not yield differences in weighted UniFrac distances (beta diversity), if the increase in alpha-diversity is the result of – for example - novel bacterial members at low relative abundance and somewhat phylogenetically related to some of the bacteria in the samples. In such a scenario, we would not expect UniFrac distances to vary significantly between week 0 and 1, while such scenario would be likely to yield an increase in alpha-diversity metrics.

Discussion

- Overall, the discussion presents a balanced synthesis of the results and is clearly written.

- Absolute and relative abundance of S. aureus declines in the nose after mupirocin

- The unexpected finding of increased bacterial abundance after mupirocin is explored and possible explanations proposed; appropriately the authors retain some uncertainty about the weight of this finding.

- In Line 337-338 the authors state “There was a sustained, although not statistically significant, disruption in the nose of those participants with a high relative abundance of S. aureus”. Looking at 5A, and the overlapping confidence intervals of the UniFrac distance, and the larger (but still not significant) disruption seen in the low abundance group, this data does not strongly support this statement, and I suggest some moderation of the findings here.

This statement was meant to reflect the fact that, in the high S. aureus group, UniFrac distances of the nose bacterial communities differed and remained somewhat constant post-mupirocin treatment compared to baseline, in comparison to the low S. aureus group where there seemed to be some “oscillations” in uniFrac distances post mupirocin treatment. We have edited this sentence (lines 349-353), which now reads: “The disruption in the nose of those participants with a high relative abundance of S. aureus, although not statistically significant, was maintained post-mupirocin treatment, in contrast to participants in the low S. aureus group where oscillations in bacterial community structure could be observed. 

Minor comments

- The abbreviation OTU (first used line 148) is not defined in the manuscript

Thank you for pointing this out. We have added text to define OTU where it is first used in the manuscript.

Reviewer #2: Thirty-two adults with staphylococcus aureus colonization were included in the study. Nose and throat specimens were collected for quantitative PCR and bacterial profiling over a 12-week period. All participants received intranasal mupirocin treatment. The conclusions are unclear.

Major revision:

Statistical methods: A mixed linear model testing for changes over time would be superior to testing for differences by repeatedly applying Wilcoxon rank sum tests. The model offers flexibility for testing several factors simultaneously. Additionally, the model can adjust for variables that were significantly different between nursing home patients and community patients as indicated in Table 1.

The reviewer makes an excellent suggestion to address the within-participant correlation using linear mixed models. We initially selected Wilcoxon tests to avoid making distributional assumptions; however, the reviewer correctly points out that these tests do not address the within-person correlation over the whole follow-up period. Since both methods have advantages and limitations, we opted to utilize both as a way to determine robustness of findings to modeling assumption. Since many variables were highly skewed and contained excessive zeroes, we used square-root transformations and using zero-inflated Gaussian linear mixed models when warranted. We considered leveraging the ability to adjust for covariates, but we did not for the following two reasons. First, source of participant (community vs. nursing home) was not the study variable in these analyses; rather participants were compared to themselves over time. However, we did observe some differences in patterns over time by S. aureus at visit 1, which led us to perform more complex modeling that accounted for time, and time-by-S. aureus interactions. Second, owing to the relatively small number of participants, we were concerned that additional adjustment (beyond the interaction terms) would lead to sparseness and conditions that would not satisfy the second-order Taylor expansion approximation to normality of the parameter-specific tests.

Minor revision:

Table 1: Indicate the statistical testing methods used to calculate the p-values.

Thank you for noticing this. We have indicated the statistical testing methods used to calculate the p-values in Table 1 in the manuscript text (lines 178-179) and added a footnote to the table.

References

1. Nouwen JL, Ott A, Kluytmans-Vandenbergh MFQ, Boelens HAM, Hofman A, van Belkum A, et al. Predicting the Staphylococcus aureus nasal carrier state: derivation and validation of a “culture rule.” Clin Infect Dis. 2004 Sep 15;39(6):806–11.

---

## [Decision Letter · Decision Letter 1]

10 May 2021

Effect of mupirocin for Staphylococcus aureus decolonization on the microbiome of the nose and throat in community and nursing home dwelling adults

PONE-D-19-32171R1

Dear Dr. Roghmann,

We’re pleased to inform you that your manuscript has been judged scientifically suitable for publication and will be formally accepted for publication once it meets all outstanding technical requirements.

Kind regards,

Giuseppe Vittorio De Socio, MD, PhD

Academic Editor

PLOS ONE

Additional Editor Comments (optional):

Reviewers' comments:

Reviewer's Responses to Questions

**Comments to the Author**

1. If the authors have adequately addressed your comments raised in a previous round of review and you feel that this manuscript is now acceptable for publication, you may indicate that here to bypass the “Comments to the Author” section, enter your conflict of interest statement in the “Confidential to Editor” section, and submit your "Accept" recommendation.

Reviewer #1: All comments have been addressed

Reviewer #2: All comments have been addressed

2. Is the manuscript technically sound, and do the data support the conclusions?

Reviewer #1: Yes

Reviewer #2: (No Response)

3. Has the statistical analysis been performed appropriately and rigorously? 

Reviewer #1: Yes

Reviewer #2: (No Response)

4. Have the authors made all data underlying the findings in their manuscript fully available?

Reviewer #1: Yes

Reviewer #2: (No Response)

5. Is the manuscript presented in an intelligible fashion and written in standard English?

Reviewer #1: Yes

Reviewer #2: (No Response)

6. Review Comments to the Author

Reviewer #1: (No further comments to authors, previous comments addressed in response)

Reviewer #2: (No Response)

7. PLOS authors have the option to publish the peer review history of their article (what does this mean?). If published, this will include your full peer review and any attached files.

Reviewer #1: No

Reviewer #2: No

---

## [Editor Report · Acceptance letter]

19 May 2021

PONE-D-19-32171R1 

Effect of mupirocin for *Staphylococcus aureus* decolonization on the microbiome of the nose and throat in community and nursing home dwelling adults 

Dear Dr. Roghmann:

I'm pleased to inform you that your manuscript has been deemed suitable for publication in PLOS ONE. Congratulations! Your manuscript is now with our production department. 

Kind regards, 

on behalf of

Dr. Giuseppe Vittorio De Socio 

Academic Editor

PLOS ONE